# M²M: Learning controllable Multi of experts and multi-scale operators are the Partial Differential Equations need

## Abstract

Learning the evolutionary dynamics of Partial Differential Equations (PDEs) is critical in understanding dynamic systems, yet current methods insufficiently learn their representations. This is largely due to the multi-scale nature of the solution, where certain regions exhibit rapid oscillations while others evolve more slowly. This paper introduces a framework of multi-scale and multi-expert (M²M) neural operators designed to simulate and learn PDEs efficiently. We employ a divide-and-conquer strategy to train a multi-expert gated network for the dynamic router policy. Our method incorporates a controllable prior gating mechanism that determines the selection rights of experts, enhancing the model's efficiency. To optimize the learning process, we have implemented a PI (Proportional, Integral) control strategy to adjust the allocation rules precisely. This universal controllable approach allows the model to achieve greater accuracy. We test our approach on benchmark 2D Navier-Stokes equations and provide a custom multi-scale dataset. M²M can achieve higher simulation accuracy and offer improved interpretability compared to baseline methods.

## 1 Introduction

Many challenges require modeling the physical world, which operates under established physical laws (Karniadakis et al., 2021; Brunton and Kutz, 2024). For example, the Navier-Stokes equations form the theoretical foundation of fluid mechanics and have widespread applications in aviation, shipbuilding, and oceanography (Vinuesa and Brunton, 2022). Various numerical approaches exist to tackle these equations. These include discretization methods such as finite difference (Godunov and Bohachevsky, 1959), finite volume (Eymard et al., 2000), finite element (Rao, 2010), and spectral methods (Shen et al., 2011). Although classical physical solvers based on first principles can achieve high accuracy, they must recalculate when faced with new problems, failing to generalize and resulting in inefficient solutions. Artificial intelligence-based surrogate models effectively address these issues by providing more adaptable and efficient solutions. Understanding and learning the data that embodies these physical laws is crucial for controlling and optimizing real-world applications (Lv et al., 2022; Kim and Boukouvala, 2020; Wang et al., 2024a). Mastery of such data-driven insights enables more precise predictions, enhanced system performance, and significant advancements in how we interact with and manipulate the application in the fields of engineering and science (Noé et al., 2020). The growing interest in efficient PDE solvers and the success of deep learning models in various fields has sparked significant attention, such as neural operator methods (Li et al., 2020; Kovachki et al., 2021; Bonev et al., 2023; Liu et al., 2024a). Um et al. (2020) proposed a spatial resolution solver to reduce the computation and accelerate physical simulations. Wu et al. (2022); Sanchez-Gonzalez et al. (2020) proposed reducing the dimensions of latent space to map the solution in the surrogate models.

*Exploring how to integrate and fully leverage performance across different scales while controlling complex learning dynamics is a promising area of research.* Based on the frequency principle (Xu et al., 2019), our primary motivation is to enable smaller or more general models to learn low-frequency dynamic data, while delegating high-frequency data to more capable models. The router (distribution policy) regulates this allocation, which sets our approach apart from other methods.

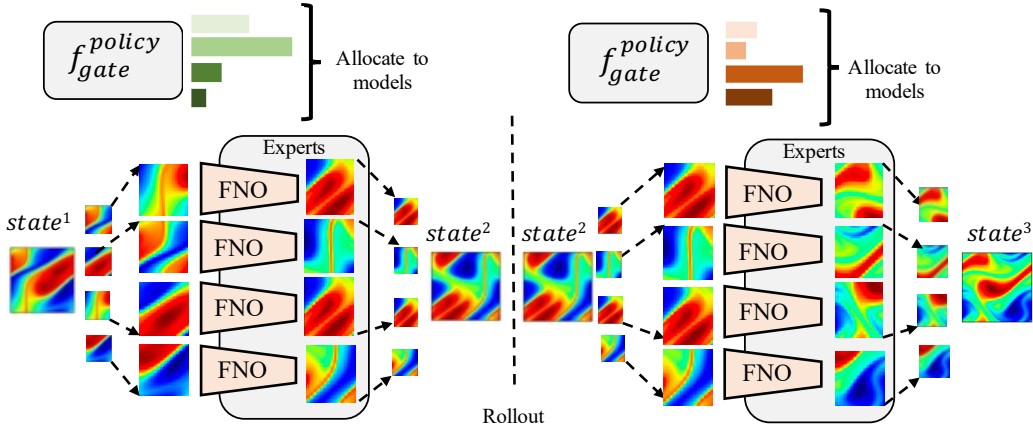

Figure 1: Framework of the proposed Multi-scale and Multi-experts (M$^2$M). The Experts net has different models, $f_{gate}^{policy}$ decides which spatial domain is needed to allocate the different models in the roll-out predications. For more details, please refer to sec. 3.

In this work, we introduce the multi-scale and multi-expert (M$^2$M) neural operators as an effective surrogate model to learn the dynamics of PDEs and optimize the appropriate allocation law for the different scales with different expert models. Our critical insight lies in leveraging the divide-and-conquer approach among models to learn the capabilities across different scales quickly. Divide and conquer is a fundamental algorithmic technique for solving complex problems by breaking them down into simpler, more manageable sub-problems (Smith, 1987; Huang et al., 2017; Ganaie et al., 2022; Emirov et al., 2024). This approach works on the principle that a large problem can often be divided into two or more smaller problems of the same or similar type. Each of these smaller problems is then solved independently. Once solutions are obtained for all the sub-problems, they are combined to form a solution to the original, more extensive problem. In addition, the model is designed to master the distribution of the most effective data while minimizing computational resources. To fairly evaluate the effectiveness of our framework, we standardized the internal models to Fourier Neural Operator (FNO) models with varying numbers of modalities. This strategy enables the model to adaptively determine the best local spatial resolution to evolve the system. The M$^2$M is trained in an alternating manner, iterating between training the evolution model with supervised loss and allocation policy net. Together, the controllable routing mechanism effectively integrates prior knowledge with model capabilities. The implementation of PID control significantly aids in optimizing the training of Multi-of-Experts (MoE).

Our main contributions are as follows:

1. We propose a controllable multi-expert and multi-scale operator model to embed multiple models based on specific priors. The multi-expert system embodies the divide-and-conquer philosophy, while the multi-scale approach enables efficient learning.

2. By bridging the control theory-PID, this unified theory demonstrates its strong generalizability. It is a versatile and scalable method for the machine learning and science simulation community.

3. We validate the aforementioned method using the standard Navier-Stokes equations and a custom multi-scale dataset, ensuring a balance between speed and accuracy.

## 2 PROBLEM SETTING AND RELATED WORK

We consider temporal Partial Differential Equations (PDEs) w.r.t. time $t \in [0, T]$ and multiple spatial dimensions $\mathbf{x} = [x_1, x_2, \ldots x_D] \in \mathbb{X} \subseteq \mathbb{R}^D$. We follow a similar notation as in (Brandstetter et al., 2022).

$$\begin{aligned}
\partial_t \mathbf{u} &= F\left(a(t), \mathbf{x}, \mathbf{u}, \mathbf{u}_x, \mathbf{u}_{xx}, \ldots\right), & (t, \mathbf{x}) &\in [0, T] \times \mathbb{X} \\
\mathbf{u}(0, \mathbf{x}) &= \mathbf{u}^0(\mathbf{x}), & \mathbf{x} &\in \mathbb{X} \\
B[\mathbf{u}](t, \mathbf{x}) &= 0, & (t, \mathbf{x}) &\in [0, T] \times \partial \mathbb{X}
\end{aligned} \tag{1}$$

where $\mathbf{u} : [0, T] \times \mathbb{X} \to \mathbb{R}^n$ is the solution, which is an infinite-dimensional function. $a(t)$ is a time-independent parameter of the system, which can be defined at each location $\mathbf{x}$, e.g. diffusion coefficient that varies in space but is static in time, or a global parameter. $F$ is a linear or nonlinear function. $\mathbf{u}^0(\mathbf{x})$ is the initial condition, and $B[\mathbf{u}](t, \mathbf{x}) = 0$ is the boundary condition when $\mathbf{x}$ is on the boundary of the domain $\partial \mathbb{X}$ across all time $t \in [0, T]$. Here $\mathbf{u}_x, \mathbf{u}_{xx}$ are first- and second-order partial derivatives, which are a matrix and a 3-order tensor, respectively (since $x$ is a vector). Solving such temporal PDEs means computing the state $\mathbf{u}(t, \mathbf{x})$ for any time $t \in [0, T]$ and location $\mathbf{x} \in \mathbb{X}$ given the above initial and boundary conditions.

The fundamental problem can be succinctly represented for tasks involving partial differential equations by the following formula.

$$(\partial\Omega, \mathbf{u}_0) \xmapsto{f_\theta} (\partial\Omega, \mathbf{u}_1) \xmapsto{f_\theta} \cdots \xmapsto{f_\theta} (\partial\Omega, \mathbf{u}_T), \tag{2}$$

where $f_\theta$ represents the model and $\partial\Omega$ denotes the boundary conditions.

**Deep Learning-based Surrogate Methods.** There are two fundamental approaches:

- Autoregressive Model Approach: The model learns the mapping function $f_\theta$ from a given $\mathbf{u}_t$ to the next $\mathbf{u}_{t+1}$, acquiring discrete representations. This method involves learning the model to predict subsequent time steps based on previous inputs. Such frameworks include CNN-based models (Wang et al., 2020b; Kemeth et al., 2022), GNN-based models (Pfaff et al., 2020; Li et al., 2024), and transformer-based models (Cao, 2021; Geneva and Zabaras, 2022; Takamoto et al., 2023).

- Neural Operator Approach: Unlike autoregressive models, the neural operator method (Lu et al., 2021) allows the model to map through multiple time steps, learning infinite-dimensional representations. This approach enables the model to handle more complex temporal dynamics by learning continuous representations. Apart from vanilla FNO, there are other operator learning methods such as U-FNO (U-Net Fourier Neural Operator, (Wen et al., 2022)), UNO (U-shaped neural operators, (Azizzadenesheli et al., 2024)), WNO (Wavelet Neural Operator, (Navaneeth et al., 2024)), and KNO (Koopman Neural Operator, (Xiong et al., 2024)).

In addition to these two conventional methods, researchers have developed several hybrid approaches that combine elements of both (Watters et al., 2017; Zhou et al., 2020; Keith et al., 2021; Hao et al., 2023; Kovachki et al., 2024; Wang et al., 2024b). For multi-scale PDEs problems, Liu et al. (2020) developed multi-scale deep neural networks, using the idea of radial scaling in the frequency domain and activation functions with compact support. Hu et al. (2023) propose the augmented physics-informed neural network (APINN), which adopts soft and trainable domain decomposition and flexible parameter sharing to further improve the extended PINN further. Xu et al. (2019) firstly find the deep neural network that fits the target functions from low to high frequencies. Liu et al. (2024b) demonstrate that for multi-scale PDEs form, the spectral bias towards low-frequency components presents a significant challenge for existing neural operators. Rahman et al. (2024) study the cross-domain attention learning method for multi-physic PDEs by the attention mechanism. However, the aforementioned methods do not efficiently leverage frequency characteristics, and they lack a controllable mechanism for adjusting the learning process of partial differential equations. Compared with (Du et al., 2023; Chalapathi et al., 2024), M$^2$M directly optimizes for the PDEs objective, first using a universal controlled method and multi-scale to learn the policy of allocating experts to achieve a better accuracy vs. computation trade-off. In addition to simplifying the computation of attention, the MoE mechanism (Jacobs et al., 1991) has been incorporated into transformer architectures (Fedus et al., 2022; Chowdhery et al., 2023) to lower computational expenses while maintaining a large model capacity. The key distinction of our objective lies in its emphasis on controllability and multi-scale considerations, both of which are crucial factors for *all fundamental partial differential equation data*.

# 3 THE PROPOSED METHOD

In this section, we detail our $M^2M$ method. We first introduce its architecture in sec. 3.1. Then we introduce its learning method (sec. 3.2), including learning objective training, and a technique to let it learn to adapt to the varying importance of error and computation. The high-level schematic is shown in figure 1.

## 3.1 MODEL ARCHITECTURE

The model architecture of $M^2M$ consists of three components: multi-scale segmentation and interpolation, Experts Net, and Gate router. We will detail them one by one.

**Multi-scale Segmentation and Interpolation.** Multi-scale segmentation involves strategically decomposing the input into multiple scales or resolutions to facilitate detailed analysis and processing. This technique benefits applications that require fine-grained analysis on various scales, such as traditional image processing (Emerson, 1998; Sunkavalli et al., 2010) and deep learning methods (Zhong et al., 2023; Yuvaraj et al., 2024). Consider a discrete form input represented as $\mathbf{u}_t^{h \times w}$, where $h$ and $w$ denote the spatial domain resolution at time step $t$. In multi-scale segmentation, the $\mathbf{u}_t^{h \times w}$ first needs to be segmented into smaller, non-overlapping scale patches. For example, segmenting a tensor $\mathbf{u}_t^{h \times w}$ into $2 \times 2$ patches results in four distinct segments. Each segment corresponds to a quarter of the original tensor, assuming that $h$ and $w$ are evenly divisible by 2. Secondly, suppose that we wish to perform an interpolation on these segmented patches to restore them to the original $h \times w$ dimensions. Mathematically, this operation can be expressed as:

$$\mathbf{u}_t^{h \times w} \xrightarrow{\text{Segmentation}} \left\{ \mathbf{u}_{i,j}^{\frac{h}{2} \times \frac{w}{2}} \Big| i,j \in \{1,2\} \right\} \xrightarrow{\text{Interpolation}} \left\{ \mathbf{P}_{i,j}^{h \times w} \Big| i,j \in \{1,2\} \right\}, \quad (3)$$

where $\mathbf{P}^{h \times w}$ represents the tensor after interpolation, which combines the four patches back into the original size of $h \times w$. This segmentation approach effectively reduces the dimensionality of each patch and allows for localized processing, which is essential for tasks involving hierarchical feature extraction.

**Experts Net.** In theory, an expert net is composed of multiple distinct models. However, our sub-expert networks are structured in a parallel configuration for rigorous comparison in this study. Importantly, we have opted for a non-hierarchical architecture. All constituent models are based on the Fourier Neural Operator (FNO), with potential variations in the number of modalities. Formally, let $E = \{E_1, E_2, \ldots, E_n\}$ represent the set of expert models, where each $E_i$ is an FNO. The input to each expert is a different patch $\mathbf{P}_i \in \mathbb{R}^{h \times w}$. The output of each expert maintains the same dimensionality as the input. The primary function of the expert system is to model the temporal evolution of the system state as shown in Eq. 2. We employ a divide-and-conquer strategy, where each expert $E_i$ operates on a subset of the input space:

$$E_i : \mathbf{P}_i^{h \times w} \rightarrow \mathbf{P'}_i^{h \times w}, \quad (4)$$

where $\mathbf{P}_i^{h \times w}$ is a patch of the input and $\mathbf{P'}_i^{h \times w}$ is the corresponding output patch. The predication solution $\hat{\mathbf{u}}_{t+1}^{h \times w}$ involves the aggregation of these individual patch predictions to reconstruct the full system state:

$$\hat{\mathbf{u}}_{t+1}^{h \times w} = A(\mathbf{P'_1}, \mathbf{P'_2}, \ldots, \mathbf{P'_n}), \quad (5)$$

where $A$ is an aggregation function that combines the individual patch predictions into a coherent global state, this approach allows for parallelization and potentially more efficient processing of complex spatio-temporal dynamics while maintaining consistency across all or sparse expert models.

**Gate Router Mechanism in MoE.** The Gate Router Mechanism is a crucial component in the MoE architecture and is responsible for distributing input patches across expert models. The primary objectives of this mechanism are:

1. To efficiently allocate different patches to different models and to route complex problems to more sophisticated networks. (Divide and Conquer)

2. To avoid overloading a single model, which could lead to high computational complexity. (Simplicity is the ultimate sophistication)

3. Optionally, the router could be set as the top-k and strong prior, which we encourage the sparse experts to apply the different regions.

Let $\mathcal{X} = \left\{ \mathbf{u}_{1,1}^{h \times w}, \mathbf{u}_{1,2}^{h \times w}, \mathbf{u}_{2,1}^{h \times w}, \mathbf{u}_{2,2}^{h \times w} \right\}$ be a set of $4$ input scale domain. The router function $R$ is defined as:

$$R : \mathcal{X} \to [0, 1]^{N \times M}, \tag{6}$$

where $R(x_i)_j$ represents the probability of routing input $x_i$ to expert $E_j$. The ideal routing strategy aims to optimize the following objectives:

$$\min_R \mathbb{E}_{x \sim \mathcal{D}} \left[ \sum_{j=1}^{N} R(x)_j \cdot \text{Error}(E_j, x) \right], \tag{7}$$

where $\mathcal{D}$ is the data distribution and $\text{Error}(E_j, x)$ is the error measure of each expert $E_j$ performs on input patch compared with the ground truth. This paper introduces an optional prior distribution $P(E_j)$ over the experts to initialize the routing mechanism and gradually train it. This prior can be incorporated into the routing decision:

$$R(x)_j = \frac{\exp(r_j(x) + \log P(E_j))}{\sum_{k=1}^{N} \exp(r_k(x) + \log P(E_k))}, \tag{8}$$

where $r_j(x)$ is a learned function that scores the suitability of expert $E_j$ for input $x$. By combining these components, the router mechanism can efficiently distribute inputs across experts, adapt to the complexity of different inputs, and maintain a balanced computational load across the system.

## 3.2 LEARNING OBJECTIVE AND CONTROL STRATEGY

The training objective is defined as follows:

$$\mathcal{L}(t) = \lambda(t) \mathcal{L}_{\text{router}} + \mathcal{L}_{\text{experts}}, \tag{9}$$

where the $\mathcal{L}_{\text{router}}$ and $\mathcal{L}_{\text{experts}}$ represent the training loss for the router and experts net, respectively. The $\lambda(t)$ is a hyperparameter related to the training epoch $t$. Our assumption is as follows: in the initial stage of the model, the router should allocate data evenly to the experts, allowing each expert to receive sufficient training. Once the expert networks have been adequately trained, if the router is not performing well, feedback should be used to train the router. This will enable the router to select the well-performing experts for further training, thereby fully leveraging the potential of the experts.

**Router Loss.** The training objective for the router can be formulated as:

$$\mathcal{L}_{\text{router}} = \text{KL}(R(x) || P(E)) + \mathcal{L}_{\text{load}}, \tag{10}$$

where KL is the Kullback-Leibler divergence, and this formulation allows the router to start from the prior distribution and gradually adapt to the optimal routing strategy as training progresses. The KL divergence term encourages the router to maintain some similarity to the prior, which can help prevent all inputs from being routed to a single expert. To promote the sparsity of the router and the computational tradeoff, we introduce a load-balancing entropy loss as $\mathcal{L}_{\text{load}}$:

$$\mathcal{L}_{\text{load}} = - \sum_{i=1}^{M} p_{ij} \log p_{ij}, \tag{11}$$

where $p_{ij}$ represents the probability $R(x_i)_j$ of assigning the $i$-th data point to the $j$-th expert.

**Expert Learning Loss.** Each expert model should be trained using supervised learning to approximate the solution of the PDE at a given time step. To achieve this, we define the loss function for each expert model using the Mean Squared Error (MSE) between the predicted solution and the true solution of the PDE at each time step.

For a given expert model $E_j$, the goal is to minimize the MSE between its prediction $\hat{u}_j(x, t)$ and the true solution $u(x, t)$ of the PDE over a set of input patches. The MSE loss for the $j$-th expert can be defined as:

$$\text{MSE}_j = \frac{1}{N} \sum_{i=1}^{N} \left( u(x_i, t_i) - \hat{u}_j(x_i, t_i) \right)^2, \tag{12}$$

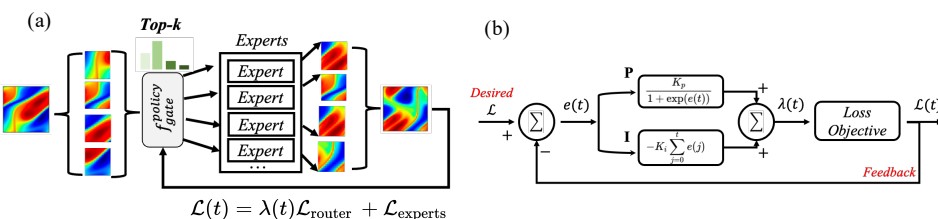

$$\mathcal{L}(t) = \lambda(t)\mathcal{L}_{\text{router}} + \mathcal{L}_{\text{experts}}$$

Figure 2: Figure (a) shows the router policy in the training. Figure (b) shows the framework of the PI controller in the M$^2$M. By designing the target and feedback in the loop, $\lambda$ can be adjusted.

where $N$ is the selected number of patches, $u(x_i, t_i)$ is the true solution at the $i$-th patch, and $\hat{u}_j(x_i, t_i)$ is the predicted solution by the expert $E_j$.

The total loss for all experts can be written as the sum of the MSE losses for each expert:

$$\mathcal{L}_{\text{experts}} = \sum_{j=1}^{M} \text{MSE}_j = \sum_{j=1}^{M} \frac{1}{N} \sum_{i=1}^{N} \left(u(x_i, t_i) - \hat{u}_j(x_i, t_i)\right)^2. \tag{13}$$

By minimizing this total loss, each expert model learns to approximate the solution of the PDE over time, ensuring that their predictions become more accurate as the training stage.

**PID-Gate Control Connects the Expert and Router.** The dispatch mechanism of the router presents a challenging issue. On the one hand, if the router initially has a strong prior, different experts may not receive sufficient training, and their specialized capabilities cannot be fully leveraged. On the other hand, if the router erroneously assigns tasks to less capable experts, the overall loss of the model may not decrease as expected. Inspired by automatic control theory (Åström et al., 2006; Wang et al., 2020a), we design a non-linear PI controller in the loop as shown in figure 2, a variant of the PID control, to automatically tune the hyperparameter $\lambda(t)$ and use the desired loss or desired prior KL distribution as feedback during model training. we also demonstrate that PID-gate control improves the performance in the ablation study. To address this challenge, we propose two control strategies in Algorithm 1, and the proof is in the Appendix 6.5.

---

**Algorithm 1** Two Dispatch Strategies with Desired Loss $\hat{L}$

---

1: **Init:** $\lambda_0, \lambda_{max}, \lambda_{min}, K_p, K_i, Top_k$, Epochs $N$, $\hat{L}$
2: **Input:** Initial PDE solution $\quad\quad\quad\quad\quad\quad\quad\quad\quad\quad\quad\quad \triangleright [Batch, T_{in}, H, W]$
3: Multiscale segmentation and interpolation $\quad\quad\quad\quad \triangleright [B, S^2, T_{in}, H, W]; S$: scale
4: **for** $t = 1$ to $N$ **do**
5: $\quad$ Router outputs probability distribution over classes
6: $\quad$ **Strategy 1:** Select top $k$ models, allocate different regions to models, and aggregate outputs with sparse models.
7: $\quad$ **Strategy 2:** Dispatch to all models, linearly combined with the weight.
8: $\quad$ **Compute Loss** $L(t)$
9: $\quad$ **Controller:** $e(t) \leftarrow L(t) - \hat{L}$; $P(t) \leftarrow \frac{K_p}{1+\exp(e(t))}$
10: $\quad$ **if** $\lambda_{min} < \lambda(t-1) < \lambda_{max}$ **then**
11: $\quad\quad$ $I(t) \leftarrow I(t-1) - K_i e(t)$
12: $\quad$ **else**
13: $\quad\quad$ $I(t) = I(t-1)$ $\quad\quad\quad\quad\quad\quad\quad\quad\quad\quad\quad\quad\quad\quad \triangleright$ Anti-wind up
14: $\quad$ **end if**
15: $\quad$ $\lambda(t) \leftarrow P(t) + I(t) + \lambda_{\min}$
16: **end for**
17: **Output:** PDE solution $\quad\quad\quad\quad\quad\quad\quad\quad\quad\quad\quad\quad \triangleright [Batch, T_{out}, H, W]$

---

## 4 EXPERIMENTS

In the following experiments, we set out to answer the following questions on our proposed M$^2$M:

- **Multi-scale effect and allocate mechanism**: Can the $M^2M$ model dynamically allocate the spatial domain to concentrate computational resources on regions with higher dynamics, thus enhancing prediction accuracy?

- **Pareto frontier improvement**: Does $M^2M$ enhance the Pareto frontier of Error versus Computation compared to deep learning surrogate models (SOTA)?

- **Controllable training**: Is $M^2M$ capable of adapting its learning results based on the dynamics of the problem, as indicated by the parameter $\lambda$?

We evaluate our $M^2M$ on two challenging datasets: (1) a custom 2D benchmark nonlinear PDEs, which tests the generalization of PDEs with the different spatial frequencies; (2) a benchmark-based Naiver-Stokes simulation generated in (Li et al., 2020). Both datasets possess multi-scale characteristics where some domains of the system are highly dynamic while others are changing more slowly. We use the relative L2 norm (normalized by ground-truth's L2 norm) as a metric, the same as in (Li et al., 2020). Since our research primarily focuses on control methods combined with multi-expert models, we aim to utilize the foundation modes of Fourier operators. In the following sections, we will consistently employ $FNO_{32}$, $FNO_{128}$, $FNO_{64}$, and $FNO_{16}$, with the goal of achieving a higher-order operator $FNO_{256}$.

### 4.1 THE COMPARISON OF CUSTOM MULTI-SCALE DATASET V.S. PID-CONTROL EFFECT

**Data and Experiments.** In this section, we test $M^2M$'s ability to balance error vs. computation tested on unseen equations with different parameters in a given family. For a fair comparison, we made the model size of the different methods as similar as possible. We use the custom dataset for testing the *Multi-scale effect* and *Controllable training*. The multi-scale dataset is given by

$$\nabla^2 u(x,y) = f(x,y), \tag{14}$$

where $u(x,y)$ is the unknown solution to be solved, and $f(x,y)$ represents the source term, which varies for different regions. More details about the multi-scale dataset are available in the Appendix 6.1.

**Main Results.** The compared baseline methods are FNO (Li et al., 2020), UNO (Azizzadenesheli et al., 2024), CNO (Raonic et al., 2024), and KNO (Xiong et al., 2024). Please refer to the appendix for baseline visualization results in the appendix 6.4.3. The $M^2M$ approach achieves Pareto optimality, as demonstrated in the Pareto frontier detailed in figure 10. As a heuristic choice, we set the target to 0 and defined the loss $L(t)$ as the RMSE in the training stage.

From figure 4, $M^2M$ can allocate models sparsely and only sends the region with a slower change (Patch number is 1-4) to lowest mode $FNO_{16}$ for the computation efficiency and highest modes of $FNO_{128}$ for accuracy both from zero-prior. The above results show that $M^2M$ can focus computation on dynamic learning in the table 4.1. $M^2M$ achieved notable improvements compared to baseline models, with Strategy 1 requiring less computation time than Strategy 2 while delivering the input to all experts. For PID-$M^2M$, we test our model on different initial $\lambda$ values and different $TOP_k$ by empirical selection of $K_p = 0.001$, $\lambda_{min} = 0$, $\lambda_{max} = 1$ and $K_i = 0.001$, where $\lambda$ focuses on router training in the initial training stage. The comparison study of the PI effect is shown in figure 3. The PI controller can speed up the error convergence in training and the value of $\lambda$ has been controlled. We see that with $\lambda$ (e.g.,

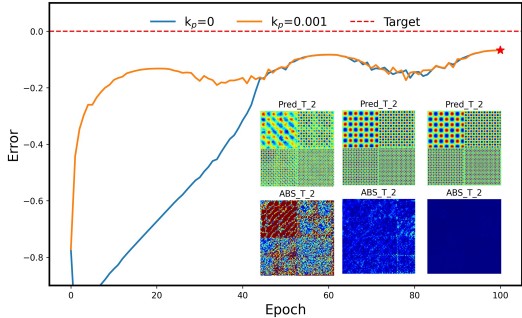

Figure 3: Results of one-step prediction on the multi-scale custom dataset at different epoch: 1, 10, and 100. The scale is set to $4$, and the ablation on the multi-scale study is shown in appendix 6.1.

$\lambda(0) = 0$ in the initial value) dynamic change, proving that the PI controller can control the router in the allocating process with different experts. To investigate whether $M^2M$ can allocate the best ex-

Table 1: Comparison study in the custom dataset. This shows that $M^2M$ can improve prediction error by selecting where to focus computation trade-offs, especially with more stringent computational constraints. All tests are conducted on an NVIDIA A800 GPU. The scale of $M^2M$ is set to 4.

| Models | | Parameters (M) | Computation (ms) | L2 error |
|---|---|---|---|---|
| **FNO** (Li et al., 2020) | $FNO_{16}$ | 0.047 | 1.9 | 0.54 |
| | $FNO_{32}$ | 0.16 | 2.2 | 0.13 |
| | $FNO_{64}$ | 0.61 | 2.3 | 0.050 |
| | $FNO_{128}$ | 2.4 | 2.5 | 0.038 |
| | $FNO_{256}$ | 9.5 | 2.6 | 0.036 |
| **UNO** (Azizzadenesheli et al., 2024) | $UNO_{16}$ | 5.2 | 6.2 | 0.080 |
| | $UNO_{32}$ | 19.2 | 8.9 | 0.075 |
| | $UNO_{64}$ | 74.2 | 18.2 | 0.042 |
| | $UNO_{128}$ | 292.6 | 33.7 | 0.026 |
| **KNO** (Xiong et al., 2024) | $KNO_{16}$ | 4.2 | 10.6 | 0.99 |
| | $KNO_{64}$ | 67.1 | 130.5 | 0.92 |
| **CNO** (Raonic et al., 2024) | $CNO_4$ | 2.0 | 18.3 | 0.12 |
| | $CNO_{64}$ | 14.2 | 148.6 | 0.010 |
| **PID-$M^2M$ (Ours)** | Strategy 1,$Top_k$=1 | 4.8 | 4.5 | 0.024 |
| | Strategy 1,$Top_k$=2 | 4.8 | 8.0 | **0.008**$^*$ |
| | Strategy 1,$Top_k$=3 | 4.8 | 11.2 | 0.012 |
| | Strategy 2 | 4.8 | 14 | **0.008**$^*$ |

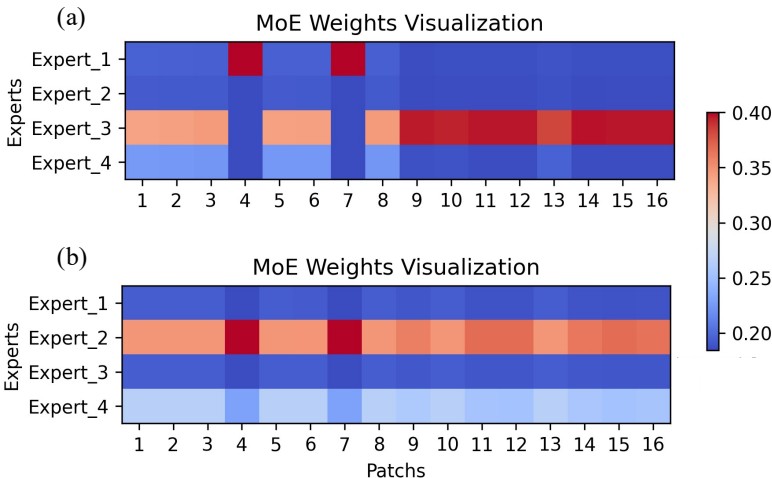

Figure 4: Dynamic weight distribution of router, the figure (a) and (b) are the distribution of the output on the 1st and 100th epoch. Prior [0000] indicates that no prior is set on the router. The $TOP_k$ is set 2.

pert on the most dynamic region according to different priors, we visualize which allocating experts on outputs of the router as shown in appendix 6.4.1.

## 4.2 THE NAIVER-STOKES (NS) DATASET AND COMPARISON OF SOTA

Here we evaluate our $M^2M$ performance in a more challenging dataset in the Naiver-Stokes dataset, the description is shown in Appendix 6.2. In this experiment, to ensure a fair comparison and leverage the $M^2M$ method's ability to enhance FNO's inherent capabilities, we selected the baseline model of FNO-3D instead of the auto-regressive style in FNO-2D. Since FNO-3D operates at least

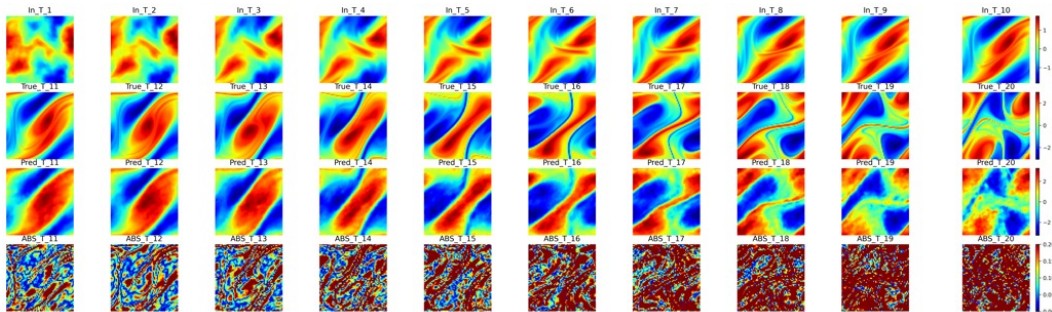

Figure 5: Results of NS datasets in PID-M$^2$M. The number of scale is 1.

three times faster than FNO-2D, this choice significantly shortened the experimental cycle. As shown in table 4.2, our M$^2$M can allocate high FNO modes to the high-frequency region and achieve better accuracy than the baselines. Specifically, M$^2$M outperforms the strong baseline of FNO$_{128}$ in the performance a little. This shows that the router could learn a proper allocation policy, allowing the evolution model to evolve the system more faithfully. However, the multi-scale effect did not perform well on this complex dataset, especially in the boundary. The reason is that there are strong temporal scale dependencies between patches, and as the spatial partitioning increases, the divide-and-conquer approach becomes less effective in the appendix 6.2. We applied our method to a cylinder wake flow in the appendix 6.4.4, close to real-world data which has the prior on the fluid mechanic. By incorporating prior distributions in regions where vortex shedding forms around the cylinder, the prediction is quite accurate.

Table 2: Comparison study in the NS dataset. This shows that M$^2$M can improve prediction error by selecting where to focus computation trade-offs, especially with more stringent computational constraints. All tests are conducted on an NVIDIA A800 GPU. The scale of M$^2$M is set to 1.

| Models | | Parameters (M) | Computation (ms) | L2 error |
|---|---|---|---|---|
| **FNO** | FNO$_{16}$ | 0.050 | 2.1 | 0.29 |
| | FNO$_{32}$ | 0.16 | 4.7 | 0.26 |
| | FNO$_{64}$ | 0.61 | 4.9 | 0.25 |
| | FNO$_{128}$ | 2.4 | 6.2 | 0.24 |
| | FNO$_{256}$ | 9.5 | 6.9 | 0.22 |
| **PID-M$^2$M** (Ours) | Strategy 1, Top$_k$=1 | 4.0 | 5.0 | 0.26 |
| | Strategy 1, Top$_k$=2 | 4.0 | 7.8 | **0.23**$^*$ |
| | Strategy 1, Top$_k$=3 | 4.0 | 13.0 | 0.25 |
| | Strategy 2 | 4.0 | 14.9 | **0.23**$^*$ |

## 5 CONCLUSION AND LIMITATION

The proposed M$^2$M model jointly learns the evolution of the physical system and optimizes computational assignment to most dynamic regions. In multi-scale and Naiver-Stokes datasets, we show that our PID method can controllably train the expert's net and router, which clearly enhances long-term prediction error than strong baselines of deep learning-based surrogate models. The fitting error has been demonstrated to converge based on control theory in the Appendix 6.5. Furthermore, the PID-based M$^2$M could improve the convergence speed, showing that this intuitive baseline is suboptimal. Finally, M$^2$M outperforms its ablation without multi-scale segmentation, showing the divide-and-conquer strategy which can significantly reduce the prediction error. We hope M$^2$M can provide valuable insight and methods for machine learning and physical simulation fields, especially for applications requiring scalability and multi-physics models. The limitation of M$^2$M is shown in appendix 6.6.

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

# 6 APPENDIX

## 6.1 CUSTOM MULTI-SCALE POISSON EQUATION DATASET

The custom multi-scale dataset is designed to simulate a complex scenario, where data in some regions change slowly while changing more rapidly in others. We define the task as follows: the input is the solution $\mathbf{u}^{low}$ to a relatively low-frequency equation, while the output is the solution $\mathbf{u}^{high}$ to a corresponding high-frequency equation. To idealize this dataset, we adopted the form of the classical Poisson equation and used the finite difference method to solve the problem. The concise discrete form is $[1, 128, 128] \mapsto [1, 128, 128]$. The time step is set to 1 and the spatial domain is set to $[128, 128]$.

### 6.1.1 FREQUENCY DISTRIBUTION ON DIFFERENT REGIONS

The source term for each region is a sinusoidal function with a systematically varying frequency. Specifically, the source term $f_{ij}(x, y)$ is defined as follows for $i, j = 1, 2, 3, 4$:

$$
\begin{aligned}
f_{11}(x, y) &= \sin(\pi \cdot (1 \cdot \mu x)) \sin(\pi \cdot (1 \cdot \mu y)), \\
f_{12}(x, y) &= \sin(\pi \cdot (2 \cdot \mu x)) \sin(\pi \cdot (2 \cdot \mu y)), \\
f_{21}(x, y) &= \sin(\pi \cdot (3 \cdot \mu x)) \sin(\pi \cdot (3 \cdot \mu y)), \\
f_{22}(x, y) &= \sin(\pi \cdot (4 \cdot \mu x)) \sin(\pi \cdot (4 \cdot \mu y)).
\end{aligned}
\tag{15}
$$

The initial solution of PDEs will be decided by the dimensionless frequency $\mu$ and the other solution for the high frequency is $7 \cdot \mu$. In this dataset, we sampled 1000 cases with different values of $\mu$, which were drawn from a normal distribution $\mathcal{N}(1, 0.1)$ using Monte Carlo sampling. Out of the 1000 samples, 700 are allocated for the training dataset, while the remaining 300 are reserved for the test dataset. To increase the complexity in the varying time PDEs, we assume that the solutions include a two-step solution and that the ground truth (the second time-solution) is the high spatial frequency to be predicted, $7 \cdot \mu$ of each low-frequency domain corresponding to the input domain.

### 6.1.2 SOLVER IMPLEMENTATION AND SETTING OF GRIDS

The Poisson equation is solved numerically using a finite-difference method on each block. The boundary conditions and the source term $f(x, y)$ determine the solution $u(x, y)$ within each block. The computational grid is set into a $128 \times 128$ grid and divided into $2 \times 2$ blocks, each of size $64 \times 64$. After calculation, the boundary condition $g(x, y) = 0$ is assigned in each block boundary.

## 6.2 2D NAIVER STOKES

### 6.2.1 2D NAIVER STOKES DATASETS

The Navier-Stokes equation has broad applications in science and engineering, such as weather forecasting and jet engine design. However, simulating it becomes increasingly challenging in the

turbulent phase, where multiscale dynamics and chaotic behavior emerge. In our work, we specifically test the model on a viscous, incompressible fluid in vorticity form within a unit torus. The concise discrete form is $[10, 64, 64] \mapsto [10, 64, 64]$. The input time step is set to 10 and the spatial domain is set to $[64, 64]$.

$$
\begin{aligned}
\partial_t w(t, x) + u(t, x) \cdot \nabla w(t, x) &= \nu \Delta w(t, x) + f(x), & x \in (0, 1)^2, \, t \in (0, T] \\
\nabla \cdot u(t, x) &= 0, & x \in (0, 1)^2, \, t \in [0, T] \\
w(0, x) &= w_0(x), & x \in (0, 1)^2
\end{aligned}
\tag{16}
$$

where $w(t, x) = \nabla \times u(t, x)$ is the vorticity, $\nu \in \mathbb{R}_+$ is the viscosity coefficient. The spatial domain is discretized into $64 \times 64$ grid. The fluid is turbulent for $\nu = 10^{-5} \left( Re = 10^5 \right)$.

### 6.2.2 RESULTS OF M$^2$M AT DIFFERENT SCALES

Figure 6 and 7 below show the prediction results at two different scales. As can be seen, there are some sharp edges at the boundaries.

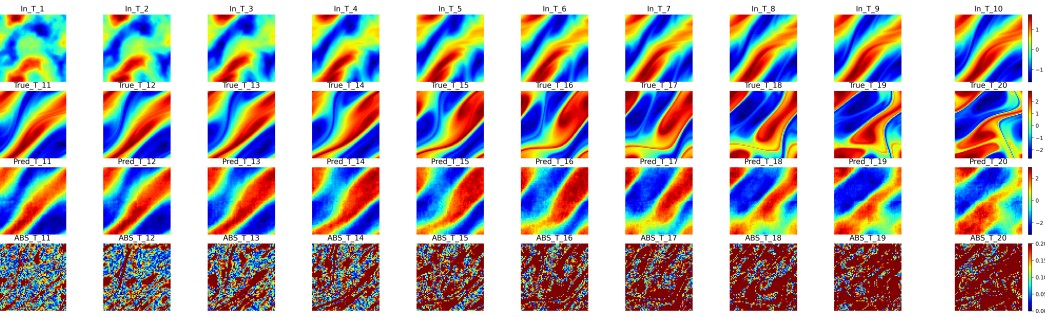

Figure 6: Model Performance at a scale factor of 2

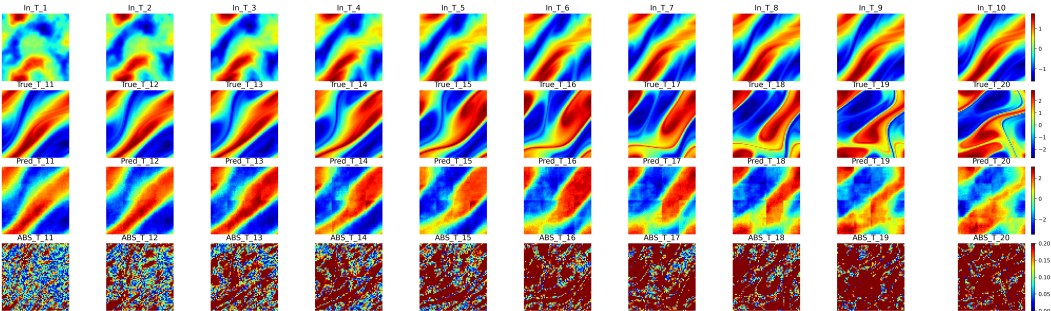

Figure 7: Model Performance at a scale factor of 4

### 6.3 DETAILED CONFIGURATION OF THE M$^2$M AND BASELINE MODELS

This section provides a detailed configuration of M$^2$M, baseline methods, and the hyperparameters used for training in Table 3 and Table 4.

### 6.3.1 HYPER-PARAMETERS FOR TRAINING OF M$^2$M

The policy network is implemented as a classifier, with the output corresponding to the weights distribution of the expert network.

Table 3: Hyperparameters for $M^2M$ architecture and training.

| Hyperparameter name | Custom dataset | 2D NS dataset |
|---|---|---|
| *Model architecture: Experts and Router* | | |
| *Experts architecture*: $FNO_{32}$, $FNO_{128}$, $FNO_{64}$, $FNO_{16}$ | | |
| *Router architecture*: Transformer based classifier | | |
| Autoregressive roll-out steps | 1 | 1 |
| Hidden channels of FNO | 6 | 6 |
| $f_{gate}^{policy}$: Transformer embedding dim | 128 | 64 |
| $f_{gate}^{policy}$: Numbers of head | 4 | 4 |
| $f_{gate}^{policy}$: Numbers of layers | 2 | 2 |
| $f_{gate}^{policy}$: Encoder layers | 2 | 2 |
| $k_p$ of PID | 0.001 | 0.001 |
| $k_i$ of PID | 0.02 | 0.02 |
| *Hyperparameters for training:* | | |
| Learning rate | $1e^{-3}$ | $1e^{-3}$ |
| Optimizer | Adam | Adam |
| Batch size | 8 | 4 |
| Number of Epochs | 100 | 200 |

### 6.3.2 HYPER-PARAMETERS FOR THE TRAINING OF BASELINE MODELS

The setting of our baseline methods is shown as follows. The hyperparameters used for training are the same as those used in the $M^2M$ model above.

Table 4: Setting for baseline models

| Hyperparameter name | Custom dataset | 2D NS dataset |
|---|---|---|
| *Baseline Operators*: FNO, UNO, KNO, and CNO | | |
| In channels | 1 | 10 |
| Modes of FNO: $16, 32, 64, 128$ | | |
| Hidden channels of FNO | 6 | 6 |
| Modes of UNO: $16, 32, 64, 128$ | | |
| Hidden channels of UNO | 6 | 6 |
| Scaling of UNO | [1,0.5,1,2,1] | [1,0.5,1,2,1] |
| Layers of UNO | 5 | 5 |
| Modes of KNO: $16, 128$ | | |
| Operator size | 6 | 6 |
| Decompose Number | 15 | 15 |
| Modes of CNO: $4, 8$ | | |
| Number of block | 4 | 4 |
| Channels | 16 | 16 |

## 6.4 EXTRA VISUALIZATION

### 6.4.1 MOE AND PID TRAJECTORY DETAILS

In this section, we present results concerning the two types of priors in the router during the initial phase, along with different PID parameters and scaling factors. One type of strong prior, such as $[0100]$ to add the output of the router, indicates that the router assigns each patch to four experts by incorporating the prior directly into the router's output through hard constraints, followed by a softmax function. The other type of weak prior represented as $[0000]$, relies entirely on the router's output without any prior constraints. As for the second prior, the results has shown in the figure 3.

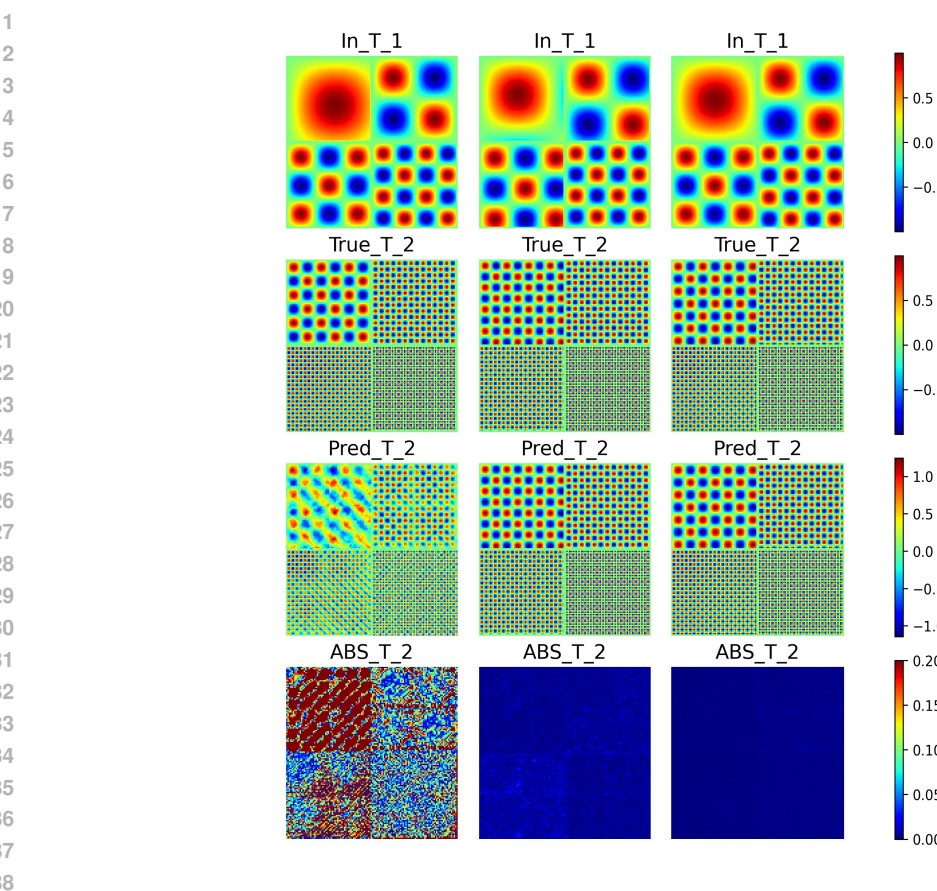

Figure 8: Variation results of input, ground truth, prediction, and absolute difference along the training epochs. The columns from left to right represent training times of 1, 50, and 100th epoch units respectively. The prior distribution for FNO is set to [0100]

### 6.4.2 ABLATION STUDY ON THE MULTI-SCALE EFFECT

We compared the performance of multi-scale models on the custom dataset, where the model is directly routed to different experts by a router, with the prior set to [0000]. It is worth noting that these comparisons were made without the inclusion of the PID algorithm, to ensure fairness in the table 5. Both interpolation and extrapolation methods in the multi-scale stage were chosen to be linear for the sake of computational efficiency.

Table 5: Ablation study on the multi-scale effect in $M^2M$. The prefix number represents the scale factor $S$, and $Top_k$ is set to 4. All tests were conducted without a controller.

| Models | RMSE | MAE |
|---|---|---|
| 1-Scale $M^2M$ | 0.015 | 0.004 |
| 2-Scale $M^2M$ | 0.010 | 0.004 |
| 4-Scale $M^2M$ | 0.008 | 0.005 |
| 8-Scale $M^2M$ | 0.008 | 0.007 |

### 6.4.3 BASELINE RESULTS

Here, we show the Pareto Frontier with different models in the figure 10. It can be observed that our $M^2M$ model, represented by the blue stars, lies on the Pareto frontier, demonstrating that our

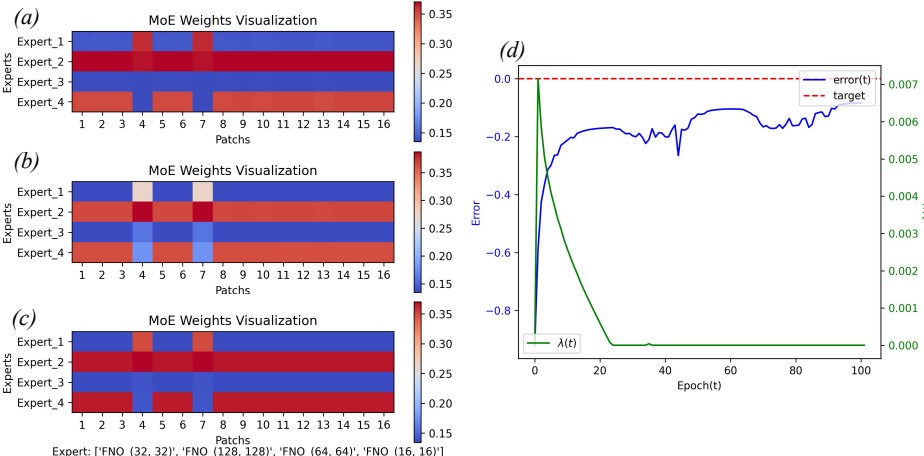

Figure 9: Variation of MoE Expert Weights with Prior and PID Parameters with scaling factor 4. Figure (a) Output of the router during the first epoch of the training stage. Figure (b) Output of the router at the 50th epoch of training. Figure (c) Output of the router at the 100th epoch. Figure (d) Adjust PID model parameters with the target set to 0. The error is defined as the difference between the model's loss function and the target. The green line represents the controllable value for $\lambda(t)$. The prior distribution for FNO is set to $[0100]$

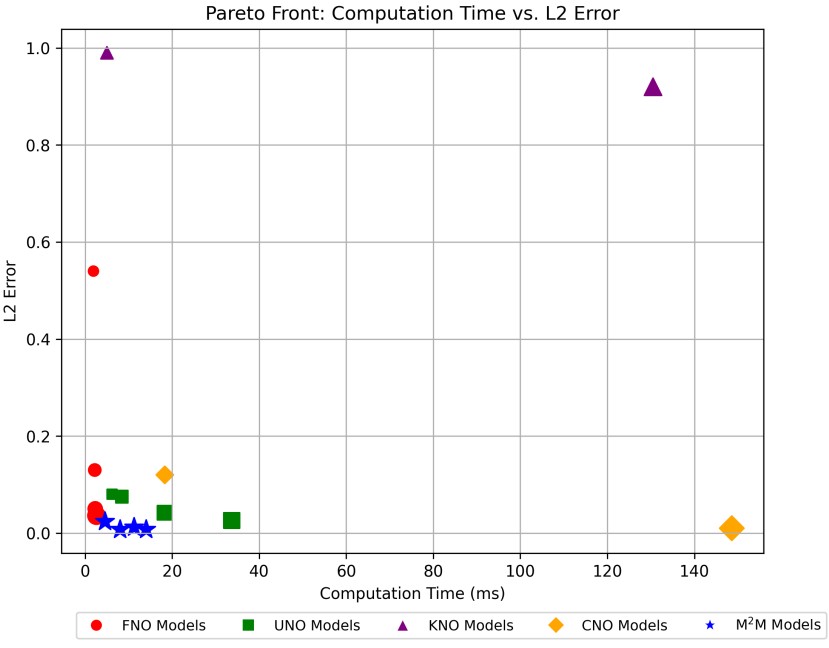

Figure 10: Pareto Frontier in the multi-scale dataset. The larger shape in the legend means larger mode numbers and larger parameters.

computational speed and accuracy are quite competitive. Performances of baseline models CNO, FNO, UNO, and KNO on the custom dataset are presented in figure 11, figure 12, figure 13, and figure 14.

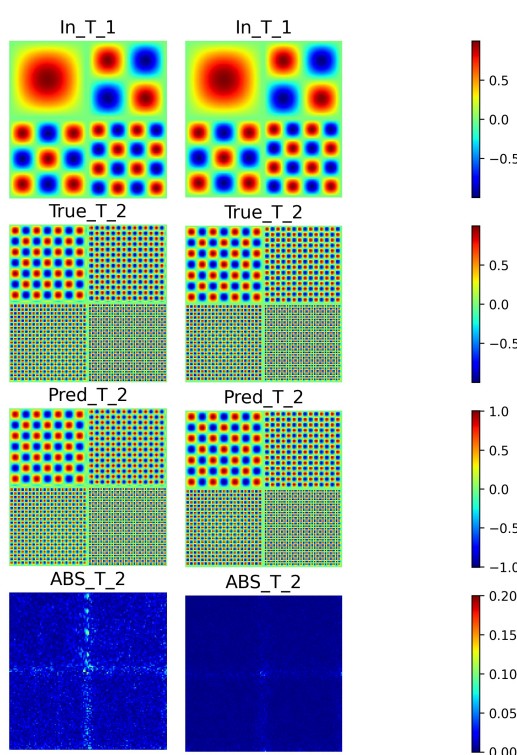

Figure 11: CNO performance on multi-scale datasets. left: $CNO_4$, right:$CNO_{16}$

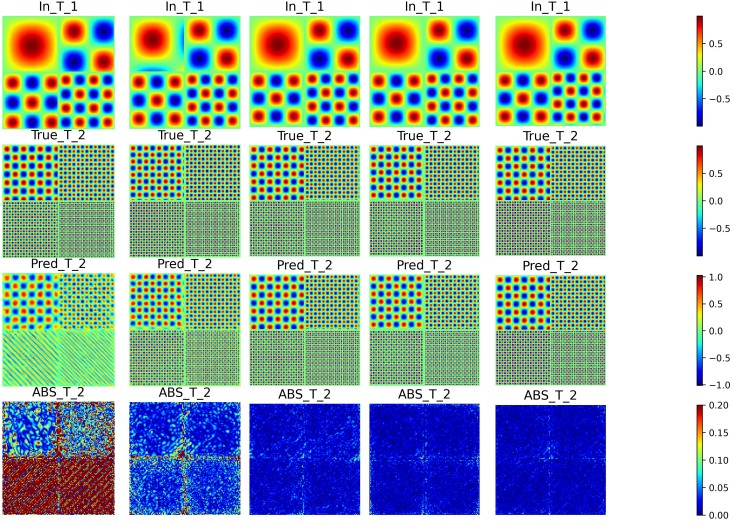

Figure 12: FNO performance on multi-scale datasets. Five Columns from left to right: $FNO_{16}$, $FNO_{32}$, $FNO_{64}$, $FNO_{128}$, and $FNO_{256}$

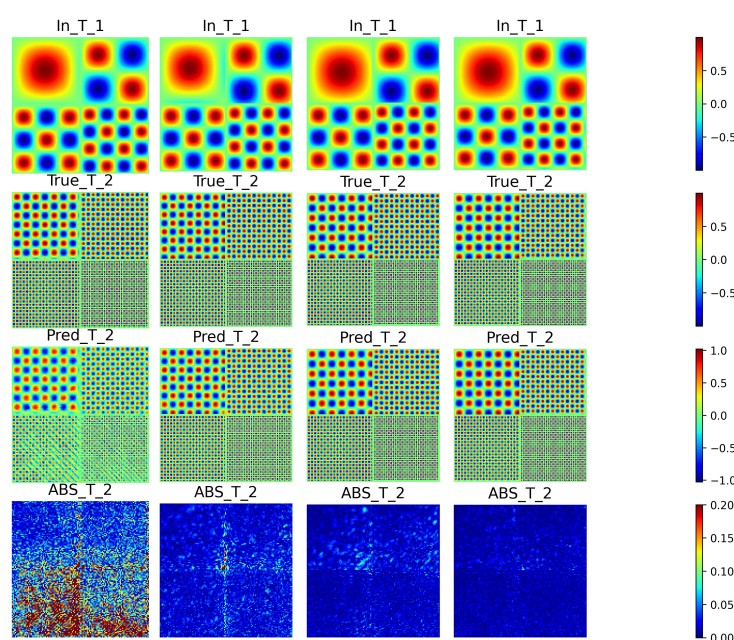

Figure 13: UNO performance on multi-scale datasets. Four Columns from left to right: $UNO_{16}$, $UNO_{32}$, $UNO_{64}$, $UNO_{128}$

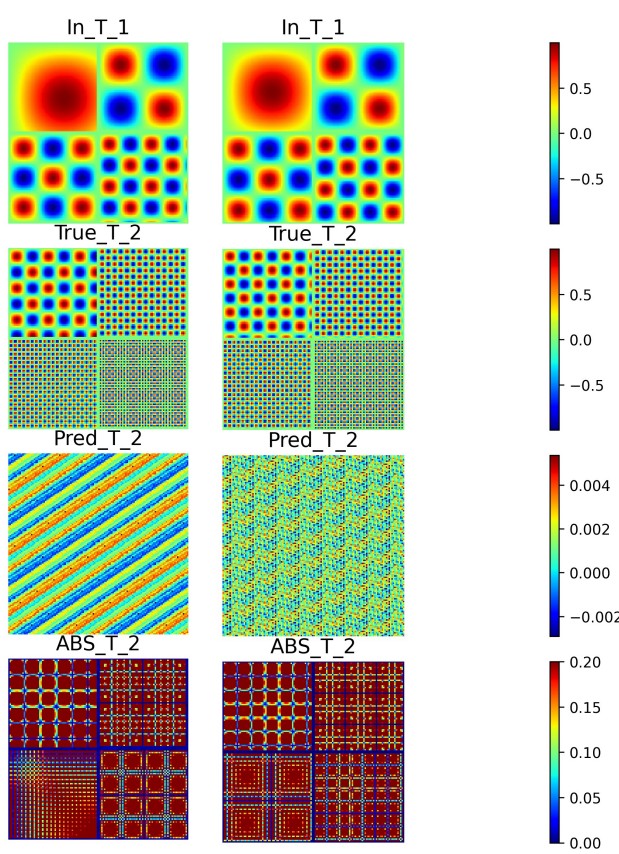

Figure 14: KNO performance on multi-scale datasets. From left to right: $KNO_{16}$, $KNO_{128}$

### 6.4.4 PERFORMANCE ON THE DATASET OF FLOWING AROUND A CYLINDER

We also consider the general application of our method on natural datasets, the dataset is shown in the following figure 15, such as the flowing around a cylinder (Tencer and Potter, 2021). The input shape $[1, 192, 112]$ needs to be mapped to the next time step with the shape $[1, 192, 112]$. The model simulation is conducted at a Reynolds number of 160, which leads to the formation of a vortex.

The allocation strong prior between different experts' models is relatively reasonable, as the vortex is generated behind the cylinder in the patch $5 - 12$. During training, it is necessary to assign the high-frequency components to the higher modes of the FNO. It is shown that $M^2M$ can learn based on the strong prior like the figure 16, which will promote artificial intelligence in scientific discoveries and simulations.

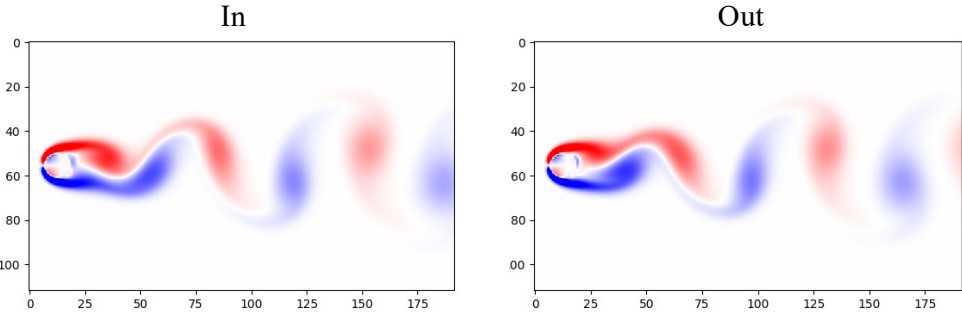

Figure 15: Dataset of flow past a cylinder, left tensor shape:$[1, 192, 112]$, right tensor shape:$[1, 192, 112]$

### 6.5 CONTROL THEORY ON PID

A PID controller regulates the control input $u(t)$ by combining three terms: proportional ($P$), integral ($I$), and derivative ($D$). The controller aims to minimize the tracking error $e(t)$, defined as the difference between the desired reference signal $r(t)$ and the system output $y(t)$:

$$e(t) = r(t) - y(t)$$

The PID control law is given by:

$$u(t) = K_p e(t) + K_i \int_0^t e(\tau)d\tau + K_d \frac{de(t)}{dt}$$

where $K_p$, $K_i$, and $K_d$ are the proportional, integral, and derivative gains, respectively.

PROOF OF CONVERGENCE

The behavior of the error $e(t)$ as $t \to \infty$. It will diminish over time due to the combined effect of the PID control effect.

1. PROPORTIONAL TERM

The proportional term $K_p e(t)$ provides an immediate response to the current error. The larger the error, the stronger the control input. This term ensures that the error decreases proportionally to its magnitude, reducing the error in time:

$$\frac{de(t)}{dt} = -K_p e(t)$$

This equation indicates that the error decreases exponentially for values of $K_p$.

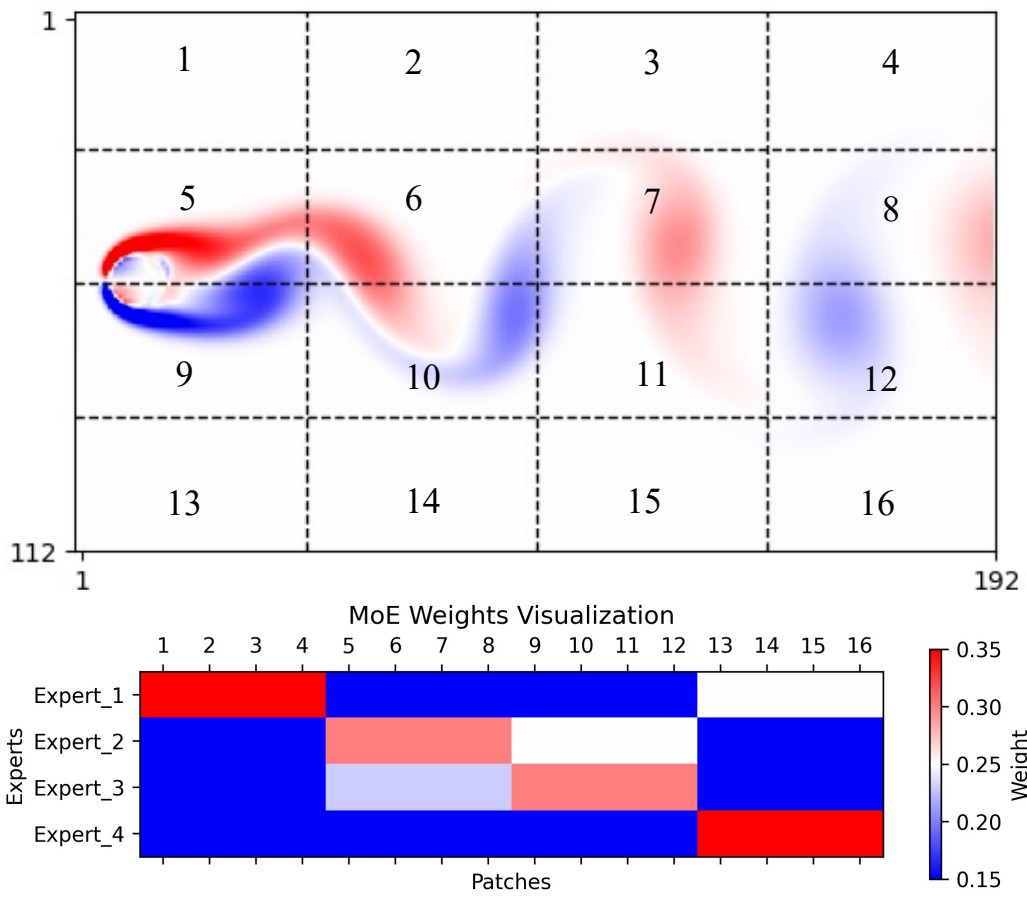

Figure 16: MOE weight on the strong prior distribution and predication result on the flow past a cylinder

2. INTEGRAL TERM

The integral term $K_i \int_0^t e(\tau)d\tau$ corrects accumulated error over time. It helps eliminate steady-state error by ensuring that small but persistent errors are corrected:

$$\frac{d}{dt}\left(\int_0^t e(\tau)d\tau\right) = e(t)$$

As long as $e(t)$ remains nonzero, the integral term grows, increasing the control input $u(t)$ until the error converges to zero.

3. DERIVATIVE TERM

The derivative term $K_d\frac{de(t)}{dt}$ anticipates future error based on the rate of change of the error. It provides a damping effect that helps reduce overshoot and oscillations in the system's response. The term is proportional to the velocity of the error, thus slowing down the system's response as the error decreases.

4. COMBINED DYNAMICS AND STABILITY

The overall system dynamics, taking all three terms into account, can be modeled as:

$$\frac{de(t)}{dt} = -K_p e(t) - K_i \int_0^t e(\tau)d\tau - K_d\frac{de(t)}{dt}$$

For a properly tuned system, the combination of the proportional, integral, and derivative terms ensures that the error will decrease over time. Specifically, the integral term guarantees that any steady-state error will be driven to zero, while the proportional and derivative terms ensure fast response and stability. Thus, as $t \to \infty$, $e(t) \to 0$.

### 6.6 LIMITATION

Our approach may have certain limitations in the following areas:

1. It may require some prior knowledge of physics, such as frequency decomposition and domain-specific knowledge embedding;

2. There may be competitive interactions between expert models, where the quality of initialization plays a decisive role in model performance;

3. The PID approach may not be suitable for more complex models and datasets, and methods like Model Predication Control or reinforcement learning might need to be explored in the future;

4. To reduce patch boundary effects, especially in scenarios where performance degrades with a high time and spatial dynamic (e.g. Turbulence) dataset;

### 6.7 MATHEMATICAL NOTATIONS

This section lists the mathematical notations used in the paper for reference.

Table 6: List of Mathematical Notations

| Symbol | Description |
|---|---|
| $\mathbf{u}$ | State variable (solution of the PDE), $\mathbf{u} : [0, T] \times \mathbb{X} \to \mathbb{R}^n$ |
| $t$ | Time variable in $[0, T]$ |
| $\mathbf{x}$ | Spatial coordinate in domain $\mathbb{X} \subseteq \mathbb{R}^D$ |
| $a$ | Time-independent parameter of the system, possibly varying with $\mathbf{x}$ |
| $F$ | Linear or nonlinear function representing PDE dynamics |
| $w(t, \mathbf{x})$ | Vorticity in the Navier-Stokes equations |
| $u(t, \mathbf{x})$ | Velocity field in the Navier-Stokes equations |
| $f(\mathbf{x})$ | Source term in PDEs |
| $\mathbb{X}$ | Spatial domain |
| $\partial \mathbb{X}$ | Boundary of the spatial domain $\mathbb{X}$ |
| $\Omega$ | Domain of the PDE problem |
| $\partial \Omega$ | Boundary of the domain $\Omega$ |
| $\mathbf{u}^0(\mathbf{x})$ | Initial condition of the PDE at $t = 0$ |
| $B[\mathbf{u}]$ | Boundary operator for boundary conditions |
| $\partial_t \mathbf{u}$ | Partial derivative of $\mathbf{u}$ with respect to time $t$ |
| $\mathbf{u_x}$ | First-order partial derivative(s) of $\mathbf{u}$ with respect to $\mathbf{x}$ |
| $\mathbf{u_{xx}}$ | Second-order partial derivative(s) of $\mathbf{u}$ with respect to $\mathbf{x}$ |
| $\nabla^2 \mathbf{u}$ | Laplacian operator (second-order differential operator of $\mathbf{u}$) |
| $f_\theta$ | Mapping function of the model, parameterized by $\theta$ |
| $E$ | Set of expert models, $E = \{E_1, E_2, \ldots, E_n\}$ |
| $E_i$ | Expert model $i$ in the multi-expert network |
| $\mathbf{P}_i$ | Segmented and interpolated patch $i$ |
| $\lambda(t)$ | Hyperparameter controlling training focus between router and experts |
| $R(\mathbf{x})_j$ | Probability of routing input $\mathbf{x}$ to expert $E_j$ |
| $p_{ij}$ | Probability of assigning data point $i$ to expert $E_j$ |
| $Top_k$ | Number of top experts selected by the router |
| $\mathcal{D}$ | Data distribution |
| $\text{Error}(E_j, \mathbf{x})$ | Error measure of expert $E_j$ on input $\mathbf{x}$ |
| $\mathcal{L}(t)$ | Total loss function at time $t$ |
| $\mathcal{L}_{\text{experts}}$ | Loss function for the experts net |
| $\mathcal{L}_{\text{router}}$ | Loss function for the router mechanism |
| $\mathcal{L}_{\text{load}}$ | Load balancing loss for the router |
| $\text{KL}(R(\mathbf{x})\|P(E))$ | Kullback-Leibler divergence between routing distribution $R(\mathbf{x})$ and prior $P(E)$ |
| $K_p, K_i, K_d$ | Proportional, Integral, and Derivative gains in the PID controller |
| $\hat{L}$ | Desired loss or target value in PID control |
| $e(t)$ | Error signal at time $t$ in PID control, $e(t) = \hat{L} - \mathcal{L}(t)$ |
| $P(t)$ | Proportional term in PID controller at time $t$, $P(t) = K_p e(t)$ |
| $I(t)$ | Integral term in PID controller at time $t$, $I(t) = I(t-1) + K_i e(t)$ |
| $\lambda_{\min}, \lambda_{\max}$ | Minimum and maximum values for $\lambda(t)$ |
| $\nu$ | Viscosity coefficient in Navier-Stokes equations |
| $Re$ | Reynolds number |
| $N$ | Number of training epochs |
| $S$ | Scale factor in multi-scale segmentation |
| $B$ | Batch size |
| $T_{\text{in}}, T_{\text{out}}$ | Number of input and output time steps |
| $H, W$ | Height and width of spatial domain grid |

