# OpenReview forum: "M$^2$M: LEARNING CONTROLLABLE MULTI OF EXPERTS AND MULTI-SCALE OPERATORS ARE THE PARTIAL DIFFERENTIAL EQUATIONS NEED"
_ICLR.cc/2025/Conference — Submitted to ICLR 2025_

### Official Review · Reviewer_ypr8 · 2024-10-31

**Soundness:** 1
**Presentation:** 1
**Contribution:** 1
**Rating:** 3
**Confidence:** 5

**Summary:**

The paper proposes a method to learn surrogates for PDE models. The proposed method builds on Fourier Neural Operators and divides the physical field to be simulated in segments. Each segment is assigned by a router to different expert models. The authors claim that their method can deal with multiple scales and multiple frequencies, offering advantages like increased flexibilty, better generalizability, more accuracy than baseline models and gain of system knowledge.

**Strengths:**

The idea to split the problem to different expert models as well as a regularization for computational complexity is interesting.

**Weaknesses:**

Motivation: The motivation is not presented well. The authors fail to adequately situate their approach within the existing body of literature. E.g. in line 14 "current methods insufficiently learn their representations" is not supported by examples or according literature. In line 50 it is claimed that "Exploring how to integrate and fully leverage performance across different scales while controlling complex learning dynamics is a promising area of research", however a reference or argumentation supporting the claim is missing.
In section 2, autoregressive and neural operators are presented as two different approaches, however the authors do not clearly position their method within both. In line 154, the authors claim “However, the aforementioned methods do not efficiently leverage frequency characteristics, and they lack a controllable mechanism for adjusting the learning process of partial differential equations.”, which is not supported by citations, experiments or further arguments. Please provide specific examples or citations from the literature that demonstrate the limitations of current methods in learning representations of multi-scale PDEs. Furthermore, references or a brief discussion of existing work on multi-scale modeling including connecntions to the proposed approach would better support their claim about the promise of this research direction and provide more structure in the literature review.

Results: The method does not increase the prediction quality for the Navier Stokes data set compared to the Fourier Neural Operator (FNO) (line  448: "Specifically, M2M outperforms the strong baseline of FNO128 in the performance a little."), which is not significant enough to support that this is achieved by the architecture design of the method. A more detailed discussion (especially on hyperparameter tuning) is missing.


The custom dataset: The custom dataset consists of fields of 4 squares with different dynamics. The proposed methodology seems to segment this in 16 squares, inducing an alignment of these squares. This is most probable not a fair comparison as 1) in physics, such quadratically structures field do not exist and 2) the alignment of dataset segments and segments that are distributed within the method induce an unfair bias to this method being superior. Experiments on publicly available datasets would provide a fairer evaluation. Usage of e.g. the PDEBench from the University of Stuttgart should be considered.

Unsupported claims: E.g. Contribution 2, Line 97 “By bridging the control theory-PID, this unified theory demonstrates its strong generalizability. It is a versatile and scalable method for the machine learning and science simulation community. “ The method is only validated for two different datasets, and a reason why PID controller integration unifies a theory is not given.
E.g. in line 442 "As shown in table 4.2, our M2M can allocate high  FNO modes to the high-frequency region" - but table 4.2 does not show this. To provide evidence on the claim of generalizability more corresponding experiments (see comment w.r.t. to datasets above) should be added. Moreover, a more detailed discussion and analysis on how the PID controller integration contributes to the method's generalizability and versatility should be added, e.g. how do the distinct controller parts (P, I and D contribute).

Clarity: The explanation of expert nets in Line 191-193 is too short and thus hard to understandable in the current status. This section should be detailed to describe the function and concept of the expert nets. Moreover, the aggregation function in eq. 6 is should be specified. Also, a more detailed discussion on the routing should be added since it is hard to follow currently. The same holds for the explanation of PI control (252, 253).  Algorithm 1 is described very imprecisely and uses not introduced variable names. According symbols and variables should be introduced and explained.

Distinction between the method presentation and its parametrization for the application is not given; this raises confusion about how the model is later used, e.g. starting in line 179, 4 patches are used, and then again n in equation 5, and then again 4 in line 219.

Graphics and tables with error metrics and terms that are not introduced/explained: e.g. caption of Figure 3 (“scale is set to 4”, how is the error metric defined), Figure 4 (“Prior [0000]”, “TOP_k”, how is the expert specified), Figure 5 (scale again, values plotted are not described, graphics quality is poor), Table 1 (what does the subsets mean).

Submitted source code is not usable: The required Python environment is not specified, there is no explanation how to use models and how to get datasets, all comments are Chinese such that they are not understandable to the worldwide community. Please provide a description or HowTo specifying the Python environment, including instructions for using the models and obtaining datasets, and translate comments to English to make the code accessible to a wider audience.

Mathematical notation unclear: use of introduced subsets (N, M, equation 6), in line 235 x misses the subscript, use of same variable for different things (e.g. x in line 107 and 222), not clear where equation 7 is used, meaning of variables in equation 10 is not clear.

Misleading introduction of PI control: The authors claim that they have a proof for the application of PI control in their method and link to the appendix. But what is claimed to be a proof is not a proof, but an introduction to PID control that can be found in many teaching books. Furthermore, intensified by the not defined error metric used in Figure 3, it remains unclear if the realized convergence increase is due to a different scaling of the components of the loss function, or a real benefit. Anyhow, to achieve the best result in Figure 3, both models still need to be trained for the same number of epochs. Additionally, the authors mismatch feedback and reference in line 297 and are not clear about what they control with PI (KL or loss?)

Paper quality: Grammar, sentence structure and spelling are frequently faulty. It seems that the quality of explanations, language and presentation gradually degrades to the end of the paper.

I think the paper introduces too many new elements of a model structure, such that the implications of the single changes cannot be separated. Here an ablation study should be added.

Important parts of results are in the appendix, as well as the limitations. This should be fixed.

Minor points:

The first two sentences in the abstract are missing arguments or citations which prove the statement “Learning the evolutionary dynamics of Partial Differential Equations (PDEs) is critical in understanding dynamic systems, yet current methods insufficiently learn their representations. This is largely due to the multi-scale nature of the solution, where certain regions exhibit rapid oscillations while others evolve more slowly.”
•	The meaning of M2M is not explained.
•	Contribution 3, line 99, is not a real contribution as validation is a must in machine learning papers “We validate the aforementioned method using the standard Navier-Stokes equations and a custom multi-scale dataset, …”
•	FNO-3D allows for zero shot super resolution, while your approach does not. This should be mentioned.
•	The title of 4.1 is not exact.
•	The numbering in the appendix is wrong; it should start with letters.

**Questions:**

Can you quantify the contribution of the different components of your architecture?

Can you provide a strong argument why your approach is supperior to the state of the art?

One major advantage of Neural Operators is being mesh independent, while the method presented seems mesh dependent. Can it be adapted to be mesh-independent?

Does your method generalize to time horizons that are longer than the trained sequence length?

How can the developments of states in the different patches interact with each other if they are always segmented in the same way?

---

### Official Review · Reviewer_5M2e · 2024-10-31

**Soundness:** 2
**Presentation:** 2
**Contribution:** 2
**Rating:** 3
**Confidence:** 5

**Summary:**

The paper, M2M...,introduces Multi-Scale and Multi-Expert neural operators, designed to simulate PDEs efficiently. The authors propose  using domain specific expert models and a divide-and-conquer strategy combined with a controllable gating mechanism that determines the allocation of computational resources across models.
The use of multiple models and the PI control strategy is innovative over standard approaches. The approach appears fresh compared to existing methods like FNO, UNO, and CNO. However, while the experimental evidence is strong, the theoretical underpinnings and the scope of generalizability could be expanded. A deeper critical evaluation of assumptions and additional testing across diverse conditions would enhance its impact.  Specifically in combining domain local solutions, considerations on boundary conditions is very critical. In multi-scale and divide-and-conquer approaches, ensuring smooth and consistent transitions between patches is critical, particularly when different models are used.  These issues are not discussed in the paper.
The dynamic allocation mechanism, while innovative, might oversimplify the complexities of real-world PDEs. The M2M framework introduces considerable architectural complexity, including multiple models, a dynamic router, and a control mechanism. While this complexity is justified in the paper as necessary for handling multi-scale problems efficiently, there is limited discussion on the practical implications, such as computational cost, scalability, and ease of implementation.
The experiments focus on a few benchmark datasets, such as custom multi-scale and Navier-Stokes simulations. While these examples are relevant, the paper does not test the model on a wider variety of PDEs or real-world scenarios that could demonstrate claims on generalizability.
Additionally the discussions in section 6. 5. on the error dynamics of the PID control related discussions is mathematically inconsistent.  Specifically
Section 4. COMBINED DYNAMICS AND STABILITY has the combined error dynamics defined such that de/dt = -u(t), this and other developments are inconsistent and incorrect with respect to control and system dynamics.
Overall I think the paper needs more work and theoretical development to provide convincing arguments toward soundness of the proposed solution.

**Strengths:**

The proposed approach appears fresh compared to existing methods like FNO, UNO, and CNO
Literature review is adequate
Paper is written well and easy to read,
Experiments are provided to support results
Supplementary material is adequate

**Weaknesses:**

The most significant weakness  is the lack of detailed explanation of how boundary conditions between segments are handled and the potential issues with inter-segment dependencies. This omission raises concerns about the physical accuracy and robustness of the method when applied to real-world problems. Coupled with the incorrect or oversimplified error dynamics analysis of the PID control law undermine the overall impact of this paper, especially for audiences that require rigorous theoretical and practical validation as is typically the case with PDE based problems that

1. Lack of Detailed Explanation on Boundary Handling. The paper does not provide any detailed explanations on how the solutions from different models or segments are combined, especially as they relate to  the handling of boundaries between segments.
2. Insufficient Theoretical Analysis, especially related to combining solutions and model selection criteria
3. Complexity considerations for practical applications and scaling, insufficient assessment to convince on broader applicability and scaling
4. Limited Scope of Experiments presented, results are based on only a  few benchmark datasets, ex.custom multi-scale and Navier-Stokes simulations. While relevant, the paper does not test the solution on a wider variety of PDEs or real-world scenarios to demonstrate generalizability
5. incorrect error dynamics for PID control. Derivations are inconsistent and appear to be derived to force asymptotic convergence to 0 error.   the derivation relates to a  very specific and uncommon case where the control input is directly designed to be the negative of the rate of change of the error, which is not typical in a PID controller setup. Instead, the error dynamics and the control input are related, but not in such a direct or simple manner.

**Questions:**

The implementation of PID control significantly aids in optimizing the training of Multi-of-Experts (MoE) -- optimizing how ?
a(t) is a time-independent parameter of the system, which can be defined at each location x, --> a(t) is time dependent not independent, explain
By bridging the control theory-PID, this unified theory demonstrates its strong generalizability -- did not demonstrate generalizability adequately
Please provide provide a specific section explaining your approach for handling boundary conditions between segments, including any techniques used to ensure continuity and consistency across segment boundaries. Also discuss potential issues that may arise from inter-segment dependencies and how their method addresses these.
Please consider including a complexity analysis section, detailing the computational and memory requirements of your approach compared to existing methods. Scaling experiments on larger problem sizes or more complex PDEs to demonstrate broader applicability will also be useful.

I am not convinced that your error dynamics developments for the PID controller are correct.  The derivations are mathematically inconsistent, please refer to one of your citations Angstorm for clarity.
Also from Fig2. your P control is Kp/(1 + exp(e(t))  ==> for large error this -->P- term =  0 ??

You may considering fixing the title,  "...MULTI OF EXPERTS '

**Details Of Ethics Concerns:**

no ethical concerns

---

### Official Review · Reviewer_YxzH · 2024-11-04

**Soundness:** 2
**Presentation:** 3
**Contribution:** 2
**Rating:** 6
**Confidence:** 2

**Summary:**

This paper introduces a framework of multi-scale and multi-expert (M2M) neural operators designed to simulate and learn PDEs efficiently.
There are three main modules in the M2M framework.  1) a divideand- conquer strategy to train a multi-expert gated network 2) a controllable prior gating mechanism and 3) a PI (Proportional, Integral) control strategy to adjust the allocation rules precisely. Experimental results on multi-scale and Naiver-Stokes datasets show the superior performance over deep learning-based surrogate models.

**Strengths:**

1. The motivation of this paper is clear.
2. The three modules make sense to me.
3. The authors have created numerous figures and visualizations to clearly illustrate the algorithm framework and experimental results.

**Weaknesses:**

1. It would be better if experiments could be conducted in more real-world scenarios to validate the results of the article.
2. There are lots of hyperparameters in the M2M method. Please give some sensitivity analysis of them.
3. The length limitation is 10 rather than 9 this year. Some important information in the Appendix can be moved to the main paper.
4. sec. -> Sec, fig. -> Fig, etc.
5. The contribution parts are not in the enumerate environment.

**Questions:**

Can the PID controller be replaced by the learning-based controller?

---

### Meta-Review · Area_Chair_HmRi · 2024-12-21

**Metareview:**

This paper proposes M$^2$M, a framework combining multi-expert and multi-scale operators for PDE dynamics, incorporating a gating mechanism and PI control strategy. While the approach shows promise with improved accuracy on limited datasets, the paper lacks clarity in its novelty compared to prior work, has a narrow experimental scope, and provides insufficient theoretical analysis of key components like the PI control strategy. Due to these limitations, I recommend rejection at this time.

**Additional Comments On Reviewer Discussion:**

During the rebuttal period, reviewers raised concerns about the paper’s unclear novelty, limited experimental validation, and lack of theoretical analysis of the proposed PI control strategy. The authors provided additional explanations to clarify their contributions and outlined plans for extending experiments in future work. However, the responses did not sufficiently address the fundamental concerns about technical novelty and comprehensive validation. These unresolved issues were significant in the final decision to recommend rejection.

---

### Decision · Program_Chairs · 2025-01-22

Reject